# Alicyclic β- and γ-Amino Acids: Useful Scaffolds for the Stereocontrolled Access to Amino Acid-Based Carbocyclic Nucleoside Analogs

**DOI:** 10.3390/molecules24010161

**Published:** 2019-01-03

**Authors:** Attila Márió Remete, Loránd Kiss

**Affiliations:** 1Institute of Pharmaceutical Chemistry, University of Szeged, Eötvös u. 6, H-6720 Szeged, Hungary; remete.attila@pharm.u-szeged.hu; 2Interdisciplinary Excellence Centre, Institute of Pharmaceutical Chemistry, University of Szeged, Eötvös u. 6, H-6720 Szeged, Hungary

**Keywords:** amino acids, nucleoside analogs, carbocycles, lactams

## Abstract

Stereocontrolled synthesis of some amino acid-based carbocyclic nucleoside analogs containing ring C=C bond has been performed on β- and γ-lactam basis. Key steps were *N*-arylation of readily available β- or γ-lactam-derived amino ester isomers and amino alcohols with 5-amino-4,6-dichloropyrimidine; ring closure of the formed adduct with HC(OMe)_3_ and nucleophilic displacement of chlorine with various *N*-nucleophiles in the resulting 6-chloropurine moiety.

## 1. Introduction and Aims

In carbocyclic nucleoside analogs, a methylene group replaces the oxygen atom in the carbohydrate ring, thereby increasing stability towards hydrolases and phosphorylases. The synthesis of these molecules is an area of considerable interest to medicinal chemistry, thanks to their bioactivity. Within natural products, neplanocin A is an antitumor antibiotic, while aristeromycin has antibacterial and antiviral activities. With respect to synthetic compounds, (-)-carbovir (**1**) and abacavir (**2**) show anti-HIV activity (Scheme 1) while entecavir inhibits the hepatitis B virus [1,2,3,4,5,6]. Carbocyclic nucleoside analogs with a 6-membered ring received less attention. In their case, antiviral activity usually requires the presence of a C=C bond in the ring [1,2,3,5,7,8] (see **3** and **4**), enabling the base to occupy a pseudoaxial position [1,5], but some (2-aminocyclohexyl)methanol derivatives (for example, **5** [9]) also exhibit bioactivity (Scheme 1).

Cyclic β-amino acids have gained significant attention in the last few decades [10,11,12,13,14]. They can be found in natural products, such as peptidyl nucleoside antibiotics amipurimycin (**6**), chryscandin (**7**), blasticidin S (**8**) or gougerotin (**9**), and related analogous derivatives (Scheme 2) [15,16,17]. In the latter three nucleoside analogs, the sugar ring was replaced with a cyclic β-amino acid unit. Cyclic β-amino acids are also promising building blocks of new bioactive peptides [14,15,16,17,18,19,20,21] and many simple representatives show relevant biological activity (Scheme 3), such as the analgesic drug tilidine (**10**) or antifungal antibiotics cispentacin and icofungipen (**11**) [10,11,12,13].

Highly-functionalized cyclic γ-amino acid derivatives possessing multiple stereogenic centers are also of considerable importance in drug research. Neuraminidase inhibitors Peramivir (**12**, Scheme 3), Zanamivir and Oseltamivir and their modified analogs are used in the treatment of influenza [22], while Gabapentin [23] and CPP-115 [24] (**13**, Scheme 3) are anticonvulsant drugs.

## 2. Results and Discussion

Taking into account the importance of carbocyclic nucleoside analogs and the bioactivity of peptidyl nucleoside antibiotics containing β-amino acids, our aim was the synthesis of new carbocyclic nucleoside analogs with an amino acid moiety on a β- and γ-lactam basis. This pathway is similar to the first synthesis of carbovir from unsaturated γ-lactam **(±)-14** (also known as Vince lactam) [25,26,27]. The synthesis of some 6-membered carbocyclic nucleoside analogs containing γ-amino alcohol was also planned.

Our synthetic work started with the opening of the heteroring of racemic Vince lactam **(±)-14** [28]. Construction of the nucleobase part on the resulting amino ester **(±)-15** was accomplished in three steps. First, compound **(±)-15** was subjected to *N*-arylation with 5-amino-4,6-dichloropyrimidine to furnish **(±)-16**. This process was accompanied by C=C bond migration thanks to the basic conditions, enabling the formation of a more stable conjugated π-system. Then, reaction with trimethyl orthoformate generated the second heteroring. The remaining chlorine atom of the obtained nucleoside analog **(±)-17** was then replaced with *N*-nucleophiles to obtain adenosine analogs **(±)-18**, **(±)-19** and **(±)-20** (Scheme 4). It is worth to note that compound **(±)-19** contains a cyclopropylamino group similar to abacavir, while the azido group of compound **(±)-20** enables many further transformations (e.g., triazole formation).

We continued our synthetic work with ethyl *cis* β-amino ester hydrochloride **(±)-22** obtained from β-lactam **(±)-21** [29,30]. Lactam ring opening, construction of the nucleobase moiety, and aromatic nucleophilic substitution resulted in nucleoside analogs **(±)-25** and **(±)-26**. From ethyl *trans* β-amino ester hydrochloride **(±)-28** [31,32], azidonucleoside **(±)-31** was prepared in a similar way (Scheme 5 and Scheme 6).

Note that the synthetic protocol took place with stereocontrol in both cases. Since the configuration of the chiral centers are not affected during the syntheses, their integrity is conserved and therefore, the *cis*-amino acid starting material led to the corresponding carbanucleoside analog in which the relative configuration of the groups is *cis*, while the *trans*-amino acid provided the carbocyclic nucleobase analog with *trans* relative steric arrangement of the ester and the heterocycle.

Analogous treatment of ethyl *cis*-2-aminocyclohex-3-enecarboxylate hydrochloride **(±)-33** (a regioisomer of **(±)-22**), obtained from β-lactam **(±)-32** [33,34], resulted in nucleoside analog **(±)-35**, the C=C regioisomer of compound **(±)-24** (Scheme 7).

In order to synthesize compounds with a five-membered carbocycle, the strategy was also extended to β-lactam **(±)-36**. Nucleoside analog **(±)-39** was obtained successfully using the protocol described above for the six-membered analogs, although both nucleobase construction steps had lower yields (Scheme 8).

Taking into account the bioactivity of compounds **3**, **(±)-4** and **(±)-5**, the synthesis of similar molecules was attempted. Reduction of β-amino acids **(±)-40** and **(±)-44** with LiAlH_4_ [32] afforded γ-amino alcohols **(±)-41** and **(±)-45**, which were further reacted with 5-amino-4,6-dichloropyrimidine. Ring closing with trimethyl orthoformate in the last step yielded, through stereocontrol, nucleoside analogs **(±)-43** and **(±)-47** (Scheme 9). These compounds show high structural similarity to bioactive compound **(±)-5**.

## 3. Conclusions and Outlook

A stereocontrolled synthetic pathway was developed to prepare new carbocyclic nucleoside analogs containing a ring olefin bond with a β-amino acid, γ-amino acid or γ-amino alcohol moiety from readily available β- and γ-lactams (across the amino acid isomers). The structure of the starting cycloalkene amino acids determined the configuration of the stereogenic centers of the products. 6- Nucleoside analogs containing the chloropurine moiety proved to be useful intermediates in various reactions with nucleophiles to access substituted nucleobases. Taking into consideration our widespread experiences in selective and controlled functionalization of versatile unsaturated cyclic amino acid derivatives [35,36,37,38], further studies in order to investigate the possible functionalization of the ring olefin bond of product nucleoside analogs are currently being investigated in our laboratory. Furthermore, based on our experiences in enzymatic resolution of various bicyclic β- and γ-lactams [39,40], as well as on enzymatic ester hydrolysis methodologies [41], synthesis of enantiomerically pure substances will be performed.

## 4. Materials and Methods

### 4.1. General Information

Chemicals were purchased from Sigma–Aldrich (Budapest, Hungary). Solvents were used as received from the suppliers. Amino ester hydrochlorides **(±)-15** [28], **(±)-22** [29,30], **(±)-28** [31,32], **(±)-33** [33,34], **(±)-37** [42] and γ-amino alcohols **(±)-41**, **(±)-45** [32] were synthesized according to literature. The ^1^H-NMR and ^13^C-NMR spectra of all new compounds are available in Appendix A.

#### 4.1.1. General Procedure for *N*-Arylation of Amino Ester Hydrochlorides with 5-Amino-2,6-Dichloropyrimidine

To a solution of the amino ester hydrochloride (10 mmoles) in EtOH (30 mL), 5-amino-2,6-dichloropyrimidine (10 mmoles) and Et3N (30 mmoles) were added, then the mixture was treated at reflux temperature for 20 h. After cooling to room temperature, the reaction mixture was concentrated under reduced pressure and the residue was taken up in EtOAc (100 mL). The organic layer was washed with water (3 × 50 mL), dried with Na_2_SO_4_, and concentrated under reduced pressure. The crude product was purified by column chromatography on silica gel (eluent: n-hexane-EtOAc 2:1).

#### 4.1.2. General Procedure for *N*-Arylation of γ-Amino Alcohols with 5-Amino-2,6-Dichloropyrimidine

To a solution of the γ-amino alcohol (8 mmoles) in EtOH (25 mL), 5-amino-2,6-dichloropyrimidine (8 mmoles) and Et_3_N (24 mmoles) were added, then the mixture was kept at boiling temperature for 20 h. After cooling to room temperature, the reaction mixture was concentrated under reduced pressure and the residue was taken up in EtOAc (100 mL). The organic layer was washed with water (3 × 40 mL), dried with Na_2_SO_4_ and concentrated under reduced pressure. The crude product was purified by column chromatography on silica gel (eluent: *n*-hexane-EtOAc 1:2).

#### 4.1.3. General Procedure for the Formation of the Purine Skeleton of Amino Ester Nucleoside Analogs

To a solution of amino ester (2 mmoles) in trimethyl orthoformate (5 mL), a catalytic amount of methanesulfonic acid or *p*-TsOH (30 mg) was added. After stirring at 20 °C for 6 h, the reaction mixture was diluted with EtOAc (25 mL) and washed with saturated aqueous NaCl solution (3 × 15 mL). The organic phase was dried with Na_2_SO_4_ and concentrated under reduced pressure. The crude product was purified by column chromatography on silica gel (eluent: *n*-hexane-EtOAc 1:1).

#### 4.1.4. General Procedure for the Formation of the Purine Skeleton of Amino Alcohol Nucleoside Analogs

To a solution of amino alcohol nucleoside analog (1 mmol) in trimethyl orthoformate (4 mL), a catalytic amount of *p*-TsOH (20 mg) was added. After stirring at 20 °C for 6 h, the reaction mixture was diluted with EtOAc (20 mL) and washed with saturated aqueous NaCl solution (3 × 15 mL). The organic phase was dried with Na_2_SO_4_ and concentrated under reduced pressure. The crude product was purified by column chromatography on silica gel (eluent: *n*-hexane-EtOAc 1:2).

#### 4.1.5. General Procedure for the Introduction of the Azido Group

To a solution of 6-chloropurinyl nucleoside analog (150 mg) in THF/H_2_O (10 mL, 4:1), sodium azide (4 eq.), acetic acid (3 drops), and Et_3_N (4 drops) were added. After heating at reflux temperature for 20 h, the reaction mixture was diluted with EtOAc (20 mL) and washed with water (2 × 15 mL). The organic phase was dried with Na_2_SO_4_ and concentrated under reduced pressure. The crude product was purified by column chromatography on silica gel (eluent: *n*-hexane-EtOAc 1:2).

#### 4.1.6. General Procedure for the Introduction of the Cyclopropylamino Group

To a solution of 6-chloropurinyl nucleoside analog (150 mg) in EtOH (10 mL), cyclopropylamine (4 eq.) was added. After the mixture was kept at boiling temperature for 12 h, the reaction mixture was concentrated under reduced pressure. The crude product was purified by column chromatography on silica gel (eluent: *n*-hexane-EtOAc 1:1).

### 4.2. Synthesis of Methylamino Compound ***(±)-18***

To a solution of 6-chloropurinyl nucleoside analogue **(±)-17** (150 mg) in EtOH (10 mL), MeNH_2_ (4 eq.) was added. After heating under reflux for 20 h, the reaction mixture was concentrated under reduced pressure. The crude product was purified by column chromatography on silica gel (eluent: *n*-hexane-EtOAc 1:1).



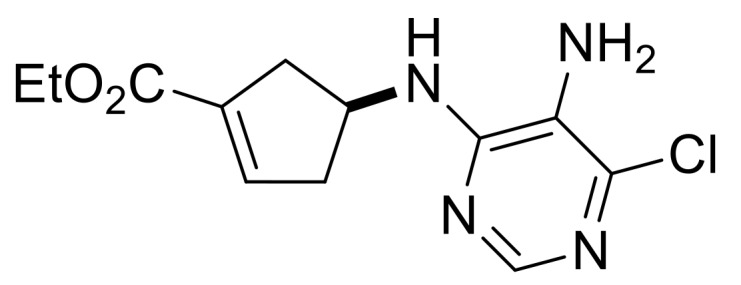

Ethyl (*S**)-4-((5-amino-6-chloropyrimidin-4-yl)amino)cyclopent-1-ene-1-carboxylate, **(±)-16**.

Brownish white solid, m.p. 121–123 °C, 40%; ^1^H-NMR (CDCl_3_, 400 MHz): *δ* (ppm) = 1.30 (t, 3H, CH_3_, *J* = 7.14 Hz), 2.43–2.59 (m, 2H, CH_2_), 3.00–3.16 (m, 2H, CH_2_), 3.44 (brs, 2H, NH_2_), 4.17–4.25 (m, 2H, OCH_2_), 4.79–4.86 (m, 1H, H-4), 5.08 (d, 1H, N-H, *J* = 5.76 Hz), 6.76-6.78 (m, 1H, H-2), 8.08 (s, 1H, Ar-H); ^13^C-NMR (DMSO, 100 MHz): *δ* (ppm) = 15.0, 39.3, 41.2, 51.3, 60.7, 124.6, 135.1, 137.7, 142.6, 146.4, 152.0, 164.9; MS (ES, pos) *m*/*z* = 283 (M + 1).



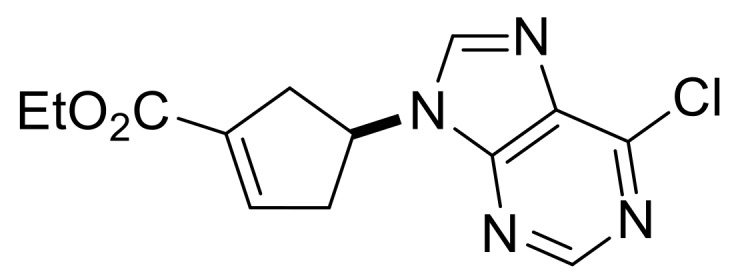

Ethyl (*S**)-4-(6-chloro-9*H*-purin-9-yl)cyclopent-1-ene-1-carboxylate, **(±)-17**.

Yellowish white solid, m.p. 83–85 °C, 81%; ^1^H-NMR (DMSO, 400 MHz): *δ* = 1.20 (t, 3H, CH_3_, *J* = 7.08 Hz), 2.93–3.05 (m, 2H, CH_2_), 3.06–3.20 (m, 2H, CH_2_), 4.09–4.17 (m, 2H, OCH_2_), 5.38–5.47 (m, 1H, H-4), 6.70–6.81 (m, 1H, H-2), 8.69 (s, 1H, Ar-H), 8.74 (s, 1H, Ar-H); ^13^C-NMR (CDCl_3_, 100 MHz): *δ* (ppm) = 14.6, 39.2, 40.9, 54.2, 61.2, 132.2, 135.5, 140.1, 143.5, 151.6, 151.8, 152.3, 164.2; MS (ES, pos) *m*/*z* = 293 (M + 1).



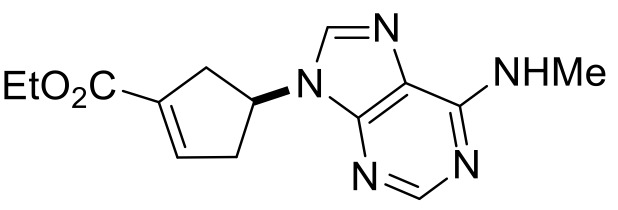

Ethyl (*S**)-4-(6-(methylamino)-9*H*-purin-9-yl)cyclopent-1-ene-1-carboxylate, **(±)-18**.

Yellow oil, 68%; ^1^H-NMR (CDCl_3_, 400 MHz): *δ* (ppm) = 1.32 (t, 3H, CH_3_, *J* = 7.12 Hz), 2.86–3.03 (m, 2H, CH_2_), 3.10–3.29 (m, 5H, NCH_3_ and CH_2_), 4.20–4.31 (m, 2H, OCH_2_), 5.41–5.50 (m, 1H, H-4), 6.16 (brs, 1H, N-H), 6.86–6.89 (m, 1H, H-2), 7.75 (s, 1H, Ar-H), 8.41 (s, 1H, Ar-H); ^13^C-NMR (CDCl_3_, 100 MHz): *δ* (ppm) = 14.6, 39.3, 41.1, 50.6, 58.9, 61.1, 120.4, 135.5, 136.1, 137.7, 140.5, 153.4, 155.9, 164.5; MS (ES, pos) *m*/*z* = 288 (M + 1).



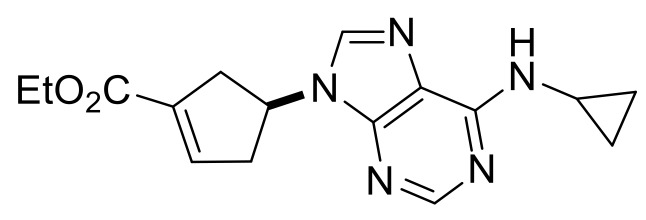

Ethyl (*S**)-4-(6-(cyclopropylamino)-9*H*-purin-9-yl)cyclopent-1-ene-1-carboxylate, **(±)-19**.

Yellow oil, 53%; ^1^H-NMR (CDCl_3_, 500 MHz): *δ* (ppm) = 0.64–0.69 (m, 2H, CH_2_), 0.90–0.97 (m, 2H, CH_2_), 1.32 (t, 3H, CH_3_, *J* = 7.13 Hz), 2.86–3.08 (m, 3H, CH_2_), 3.18–3.32 (m, 2H, CH_2_, CH), 4.21–4.28 (m, 2H, OCH_2_), 5.35–5.43 (m, 1H, H-4), 6.03 (brs, 1H, N-H), 6.86-6.91 (m, 1H, H-2), 7.75 (s, 1H, Ar-H), 8.48 (s, 1H, Ar-H); ^13^C-NMR (CDCl_3_, 126 MHz): *δ* (ppm) = 7.4, 14.3, 23.7, 38.9, 40.8, 52.9, 60.7, 119.9, 135.1, 137.6, 140.2, 153.2, 155.8, 164.1; MS (ES, pos) *m*/*z* = 314 (M + 1).



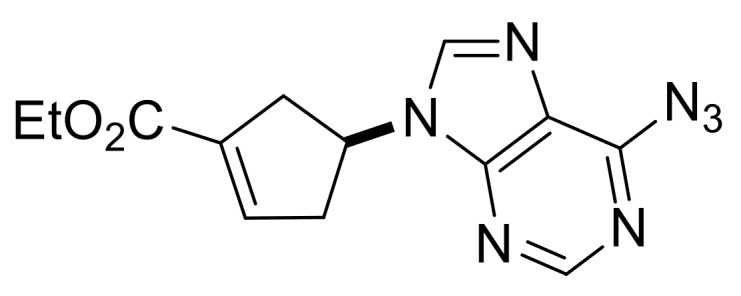

Ethyl (*S**)-4-(6-azido-9*H*-purin-9-yl)cyclopent-1-ene-1-carboxylate, **(±)-20**.

White solid, m.p. 145–147 °C, 68%; ^1^H-NMR (DMSO, 400 MHz): *δ* (ppm) = 1.22 (t, 3H, CH_3_, *J* = 7.08 Hz), 2.97–3.09 (m, 2H, CH_2_), 3.22–3.28 (m, 2H, CH_2_), 4.12–4.23 (m, 2H, OCH_2_), 5.53–5.64 (m, 1H, H-4), 6.86–6.89 (m, 1H, H-2), 8.66 (s, 1H, Ar-H), 10.07 (s, 1H, Ar-H); ^13^C-NMR (DMSO, 100 MHz): *δ* (ppm) = 15.0, 34.9, 39.2, 55.1, 61.0, 121.2, 134.5, 136.3, 141.8, 142.8, 143.8, 146.3, 164.5; MS (ES, pos) *m*/*z* = 300 (M + 1).



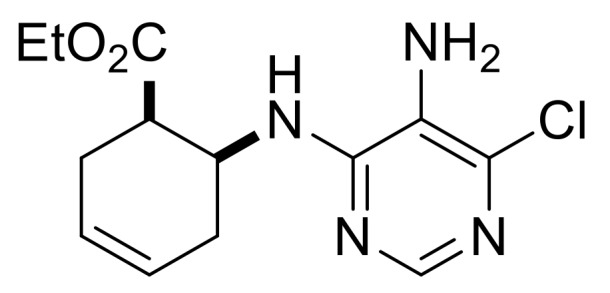

Ethyl (1*R**,6*S**)-6-((5-amino-6-chloropyrimidin-4-yl)amino)cyclohex-3-ene-1-carboxylate, **(±)-23**.

White solid, m.p. 120–121 °C, 42%; ^1^H-NMR (CDCl_3_, 400 MHz): *δ* (ppm) = 1.28 (t, 3H, CH_3_, *J* = 7.12 Hz), 2.26–2.38 (m, 1H, CH_2_), 2.42–2.56 (m, 2H, CH_2_), 2.62–2.73 (m, 1H, CH_2_), 2.95–3.03 (m, 1H, H-1), 3.39 (brs, 2H, NH_2_), 4.11–4.24 (m, 2H, OCH_2_), 4.74–4.85 (m, 1H, H-6), 5.66–5.87 (m, 3H, H-3, H-4, N-H), 8.08 (s, 1H, Ar-H); ^13C^-NMR (DMSO, 100 MHz): *δ* (ppm) = 14.8, 25.5, 30.4, 41.4, 47.1, 60.6, 124.6, 125.4, 125.9, 138.0, 146.1, 152.2, 173.4; MS (ES, pos) *m*/*z* = 297 (M + 1), 299 (M + 3).



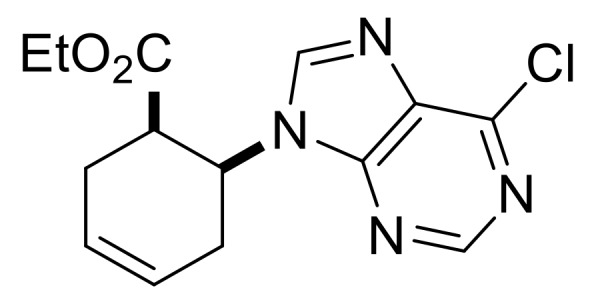

Ethyl (1*R**,6*S**)-6-(6-chloro-9*H*-purin-9-yl)cyclohex-3-ene-1-carboxylate, **(±)-24**.

Yellow oil, 76%. ^1^H-NMR (DMSO, 400 MHz): *δ* (ppm) = 0.98 (t, 3H, CH_3_, *J* = 7.12 Hz), 2.31–2.52 (m, 2H, CH_2_), 2.74–2.84 (m, 2H, CH_2_), 3.28–3.33 (m, 1H, H-1), 3.83–3.90 (m, 2H, OCH_2_), 5.25–5.32 (m, 1H, H-6), 5.85–5.89 (m, 2H, H-3, H-4), 8.56 (s, 1H, Ar-H), 8.78 (s, 1H, Ar-H). ^13^C-NMR (DMSO, 100 MHz): *δ* (ppm) = 14.5, 25.4, 29.3, 42.0, 51.2, 61.2, 125.3, 126.4, 131.2, 146.6, 149.9, 152.2, 153.0, 172.4; MS (ES, pos) *m*/*z* = 307 (M + 1).



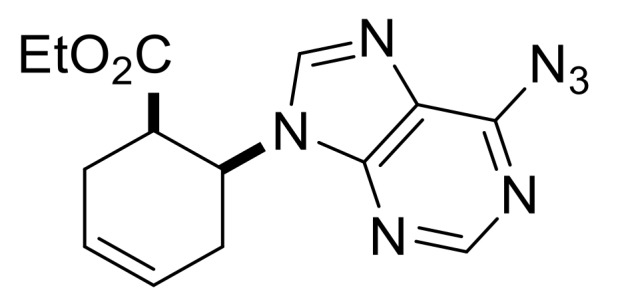

Ethyl (1*R**,6*S**)-6-(6-azido-9*H*-purin-9-yl)cyclohex-3-ene-1-carboxylate, **(±)-25**.

White solid, m.p. 130–132 °C, 51%; ^1^H-NMR (DMSO, 400 MHz): *δ* (ppm) = 1.03 (t, 3H, CH_3_, *J* = 7.10 Hz), 2.34–2.65 (m, 2H, CH_2_), 2.78–3.00 (m, 2H, CH_2_), 3.31–3.43 (m, 1H, H-1), 3.82–4.00 (m, 2H, OCH_2_), 5.40-5.48 (m, 1H, H-6), 8.54 (s, 1H, Ar-H), 10.13 (s, 1H, Ar-H); ^13^C-NMR (DMSO, 100 MHz): *δ* (ppm) = 14.6, 25.4, 29.7, 42.3, 51.5, 61.3, 125.3, 126.4, 129.4, 130.4, 135.3, 142.8, 147.9, 170.5; MS (ES, pos) *m*/*z* = 314 (M + 1).



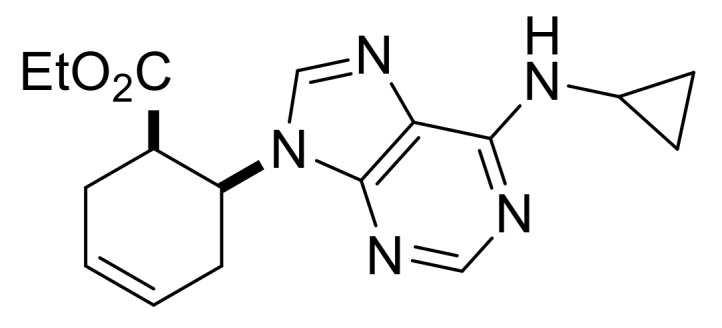

Ethyl (1*R**,6*S**)-6-(6-(cyclopropylamino)-9*H*-purin-9-yl)cyclohex-3-ene-1-carboxylate, **(±)-26**.

White solid, m.p. 128–129 °C, 62%; ^1^H-NMR (CDCl_3_, 400 MHz): *δ* (ppm) = 0.61–0.65 (m, 2H, CH_2_), 0.88–0.93 (m, 2H, CH_2_), 1.20 (t, 3H, CH_3_), 2.48–2.55 (m, 2H, CH_2_), 2.70–2.74 (m, 2H, CH_2_), 2.99–3.04 (m, 1H, H-1), 3.14–3.20 (m, 1H, CH), 3.97–4.08 (m, 2H, OCH_2_), 5.30–5.38 (m, 1H, H-6), 5.91–5.98 (m, 2H, H-3, H-4), 6.04 (brs, 1H, N-H), 7.98 (s, 1H, Ar-H), 8.44 (s, 1H, Ar-H); ^13^C-NMR (CDCl_3_, 126 MHz): *δ* (ppm) = 7.4, 13.9, 23.7, 25.2, 29.8, 42.0, 49.2, 61.0, 119.3, 124.8, 125.9, 139.0, 148.6, 153.0, 155.7, 172.1; MS (ES, pos) *m*/*z* = 328 (M + 1).



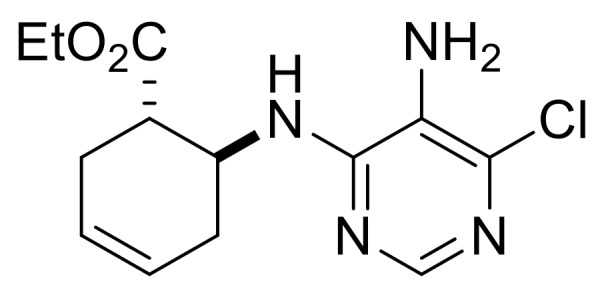

Ethyl (1*S**,6*S**)-6-((5-amino-6-chloropyrimidin-4-yl)amino)cyclohex-3-ene-1-carboxylate, **(±)-29**.

White solid, m.p. 120–121 °C, 42%; ^1^H-NMR (CDCl_3_, 400 MHz): *δ* (ppm) = 1.19 (t, 3H, CH_3_, *J* = 7.12 Hz), 2.03–2.12 (m, 1H, CH_2_), 2.34–2.43 (m, 1H, CH_2_), 2.56–2.75 (m, 2H, CH_2_), 2.83–2.92 (m, 1H, H-1), 3.47 (brs, 2H, NH_2_), 4.06–4.16 (m, 2H, OCH_2_), 4.62–4.69 (m, 1H, H-6), 5.13 (d, 1H, N-H, *J* = 8.12 Hz), 5.66–5.78 (m, 2H, H-3, H-4), 8.09 (s, 1H, Ar-H); ^13C^-NMR (CDCl_3_, 100 MHz): *δ* (ppm) = 14.5, 27.5, 31.9, 45.2, 48.5, 61.3, 122.1, 124.2, 125.1, 143.7, 150.1, 154.8, 174.1; MS (ES, pos) *m*/*z* = 297 (M + 1).



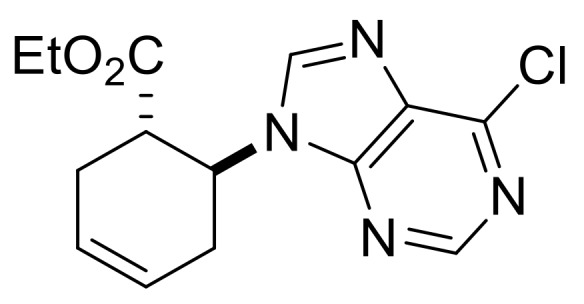

Ethyl (1*S**,6*S**)-6-(6-chloro-9H-purin-9-yl)cyclohex-3-ene-1-carboxylate, **(±)-30**.

Colorless oil, 58%; ^1^H-NMR (CDCl_3_, 400 MHz): *δ* (ppm) = 0.96 (t, 3H, CH_3_, *J* = 7.12 Hz), 2.53–2.68 (m, 3H, CH_2_), 3.01–3.07 (m, 1H, CH_2_), 3.56–3.66 (m, 1H, H-1), 3.86–3.96 (m, 2H, OCH_2_), 4.94–5.03 (m, 1H, H-6), 5.78−5.89 (m, 2H, H-3, H-4), 8.17 (s, 1H, Ar-H), 8.76 (s, 1H, Ar-H); ^13^C-NMR (CDCl_3_, 100 MHz): *δ* (ppm) = 14.2, 29.0, 30.7, 44.1, 54.5, 61.4, 124.4, 125.6, 132.2, 145.3, 151.5, 151.9, 152.1, 173.0; MS (ES, pos) *m*/*z* = 307 (M + 1).



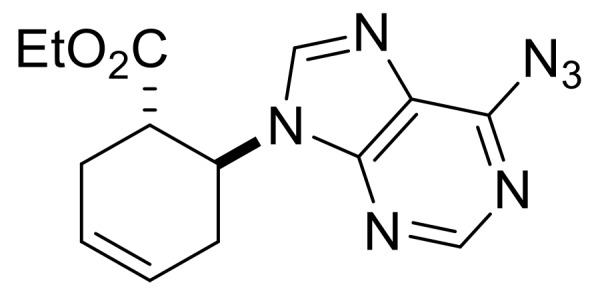

Ethyl (1*S**,6*S**)-6-(6-azido-9*H*-purin-9-yl)cyclohex-3-ene-1-carboxylate, **(±)-31**.

White solid, m.p. 140–141 °C, 68%; ^1^H-NMR (DMSO, 400 MHz): *δ* (ppm) = 0.74 (t, 3H, CH_3_, *J* = 7.08 Hz), 2.46–2.57 (m, 3H, CH_2_), 2.86–3.00 (m, 1H, CH_2_), 3.60–3.66 (m, 1H, H-1), 3.69–3.76 (m, 2H, OCH_2_), 5.08–5.15 (m, 1H, H-6), 5.80–5.92 (m, 2H, H-3, H-4), 8.79 (s, Ar-H), 10.13 (s, 1H, Ar-H). ^13^C-NMR (DMSO, 100 MHz): *δ* (ppm) = 14.3, 29.3, 31.9, 44.7, 54.3, 61.1, 125.1, 125.9, 136.5, 139.3, 142.9, 144.6, 146.3, 173.1. MS (ES, pos) *m*/*z* = 314 (M + 1).



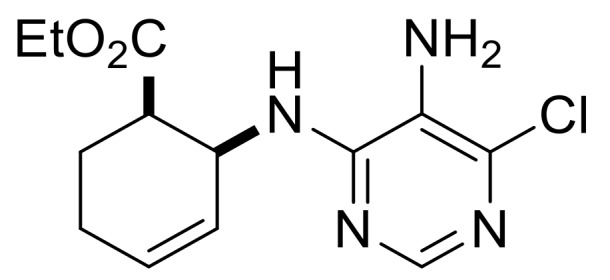

Ethyl (1*R**,2*S**)-2-((5-amino-6-chloropyrimidin-4-yl)amino)cyclohex-3-ene-1-carboxylate, **(±)-34**.

Brown oil, 55%; ^1^H-NMR (CDCl_3_, 400 MHz): *δ* (ppm) = 1.20 (t, 3H, CH_3_, *J* = 7.14 Hz), 1.99–2.19 (m, 4H, CH_2_), 2.99–3.05 (m, 1H, H-1), 4.00–4.17 (m, 2H, OCH_2_), 5.23–5.26 (m, 1H, H-2), 5.59 (d, 1H, N-H, *J* = 9.04 Hz), 5.70–5.78 (m, 1H, H-4), 5.87–5.92 (m, 1H, H-3), 8.06 (s, 1H, Ar-H), ^13C^-NMR (CDCl_3_, 100 MHz): *δ* (ppm) = 14.5, 22.7, 23.6, 43.4, 46.9, 61.1, 122.4, 127.5, 130.0, 143.4, 149.7, 154.6, 174.2; MS (ES, pos) *m*/*z* = 297 (M + 1).



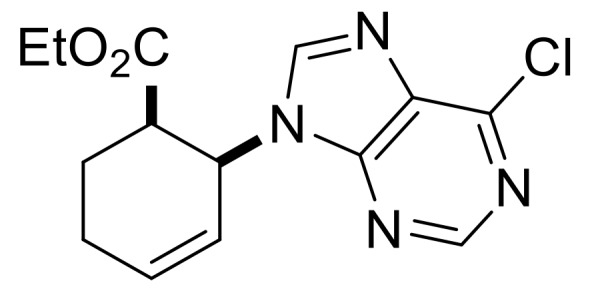

Ethyl (1*R**,2*S**)-2-(6-chloro-9*H*-purin-9-yl)cyclohex-3-ene-1-carboxylate, **(±)-35**.

Brown solid, m.p. 119–121 °C, 68%; ^1^H-NMR (CDCl_3_, 400 MHz): *δ* (ppm) = 1.03 (t, 3H, CH_3_, *J* = 7.14 Hz), 2.00–2.08 (m, 2H, CH_2_), 2.27–2.33 (m, 1H, CH_2_), 2.41–2.46 (m, 1H, CH_2_), 3.12–3.20 (m, 1H, H-1), 3.72–3.87 (m, 2H, OCH_2_), 5.67–5.70 (m, 1H, H-2), 5.83–5.90 (m, 1H, H-4), 6.28–6.33 (m, 1H, H-3), 8.23 (s, 1H, Ar-H), 8.73 (s, 1H, Ar-H); ^13^C-NMR (DMSO, 100 MHz): *δ* (ppm) = 14.3, 20.3, 24.4, 44.1, 50.1, 61.0, 123.1, 134.8, 147.5, 147.8, 149.9, 152.3, 152.9, 172.4; MS (ES, pos) *m*/*z* = 307 (M + 1), 309 (M + 3).



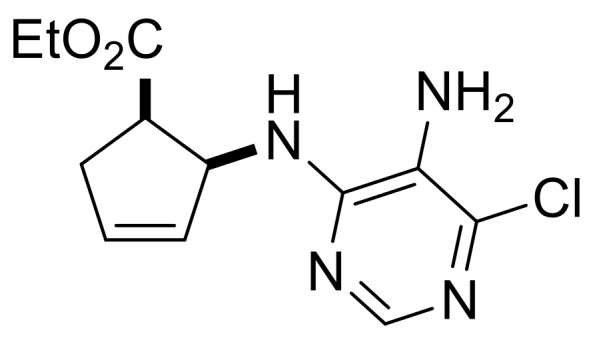

Ethyl (1*R**,2*S**)-2-((5-amino-6-chloropyrimidin-4-yl)amino)cyclopent-3-ene-1-carboxylate, **(±)-38**.

Brownish white solid, m.p. 104–106 °C, 34%; ^1^H-NMR (CDCl_3_, 400 MHz): *δ* (ppm) = 1.04 (t, 3H, CH_3_, *J* = 7.12 Hz), 2.56–2.67 (m, 1H, CH_2_), 2.83–2.90 (m, 1H, CH_2_), 3.44–3.54 (m, 1H, H-1), 3.57 (brs, 2H, NH_2_), 3.85–4.02 (m, 2H, OCH_2_), 5.34 (d, 1H, N-H, *J* = 8.76 Hz), 5.69–5.76 (m, 2H, H-2, H-4), 5.98–6.01 (m, 1H, H-3), 8.06 (s, 1H, Ar-H); ^13^C-NMR (CDCl_3_, 100 MHz): *δ* (ppm) = 14.2, 35.4, 46.4, 58.2, 61.1, 122.6, 130.2, 134.0, 143.0, 149.3, 154.1, 174.0; MS (ES, pos) *m*/*z* = 283 (M + 1), 285 (M + 3).



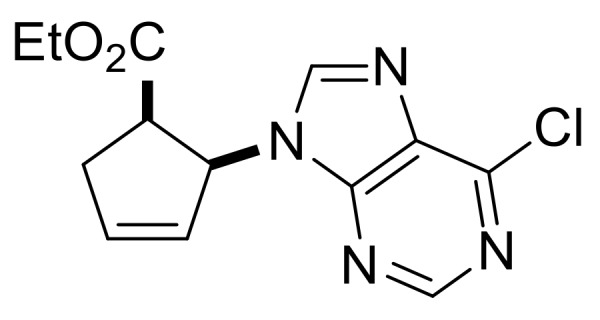

Ethyl (1*R**,2*S**)-2-(6-chloro-9*H*-purin-9-yl)cyclopent-3-ene-1-carboxylate, **(±)-39**.

Yellowish white solid, m.p. 118–119 °C, 39%; ^1^H-NMR (CDCl_3_, 400 MHz): *δ* (ppm) = 0.72 (t, 3H, CH_3_, *J* = 7.14 Hz), 2.76–2.85 (m, 1H, CH_2_), 3.14–3.23 (m, 1H, CH_2_), 3.49–3.56 (m, 1H, H-1), 3.65–3.77 (m, 2H, OCH_2_), 5.86–5.89 (m, 1H, H-2), 5.15–5.17 (m, 1H, H-4), 6.44–6.47 (m, 1H, H-3), 7.99–8.01 (m, 1H, Ar-H), 8.78–8.81 (m, 1H, Ar-H); ^13^C-NMR (CDCl_3_, 100 MHz): *δ* (ppm) = 13.8, 34.9, 47.2, 60.9, 61.4, 126.8, 134.7, 138.7, 150.0, 152.2, 152.3, 154.2, 170.9; MS (ES, pos) *m*/*z* = 293 (M + 1), 295 (M + 3).



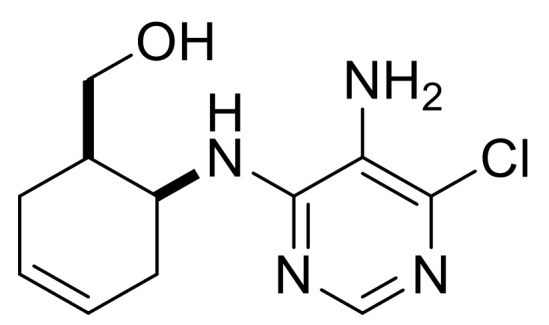

((1*R**,6*S**)-6-((5-Amino-6-chloropyrimidin-4-yl)amino)cyclohex-3-en-1-yl)methanol, **(±)-42**.

White solid, m.p. 186–188 °C, 60%; ^1^H-NMR (DMSO, 400 MHz): *δ* (ppm) = 1.96–2.12 (m, 4H, CH_2_), 2.21–2.30 (m, 1H, H-1), 3.24–3.32 (m, 1H, OCH_2_), 3.40-3.49 (m, 1H, OCH_2_), 4.43–4.48 (m, 2H, H-6 and O-H), 5.16 (brs, 2H, N-H), 5.59–5.70 (m, 2H, H-3, H-4), 6.24 (d, 1H, N-H, *J* = 7.56 Hz), 7.68 (s, 1H, Ar-H); ^13^C-NMR (DMSO, 100 MHz): *δ* (ppm) = 25.9, 30.1, 39.8, 47.2, 61.4, 124.5, 125.6, 126.6, 138.0, 146.4, 152.6; MS (ES, pos) *m*/*z* = 255 (M + 1), 257 (M + 3).



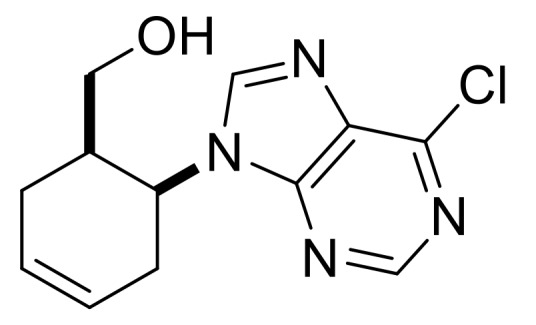

((1*R**,6*S**)-6-(6-Chloro-9*H*-purin-9-yl)cyclohex-3-en-1-yl)methanol, **(±)-43**.

White solid, m.p. 109–111 °C, 76%, ^1^H-NMR (CDCl_3_, 500 MHz): *δ* (ppm) = 1.36–1.47 (m, 1H, CH_2_), 2.05–2.15 (m, 1H, CH_2_), 2.34–2.44 (m, 1H, H-1), 2.50–2.59 (m, 1H, CH_2_), 2.73–2.81 (m, 1H, OCH_2_), 2.93–3.03 (m, 1H, CH_2_), 3.44–3.53 (m, 1H, OCH_2_), 4.65 (brs, 1H, OH), 5.31–5.37 (m, 1H, H-6), 5.98–6.06 (m, 2H, H-3, H-4), 8.33 (s, 1H, Ar-H), 8.76 (s, 1H, Ar-H); ^13^C-NMR (CDCl_3_, 126 MHz): 22.6, 31.0, 39.7, 48.6, 62.1, 124.7, 127.8, 131.1, 144.6, 151.6, 151.7, 152.5, MS (ES, pos) *m*/*z* = 265 (M + 1), 267 (M + 3).



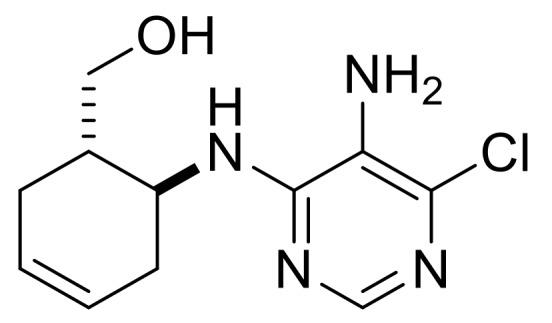

((1*S**,6*S**)-6-((5-Amino-6-chloropyrimidin-4-yl)amino)cyclohex-3-en-1-yl)methanol, **(±)-46**.

White solid, m.p. 164–167 °C, 64%; ^1^H-NMR (DMSO, 400 MHz): *δ* (ppm) = 1.70–1.84 (m, 1H, CH_2_), 1.89–2.05 (m, 2H, CH_2_), 2.14–2.25 (m, 1H, CH_2_), 2.26–2.36 (m, 1H, H-1), 3.26–3.43 (m, 2H, OCH_2_), 4.05–4.15 (m, 1H, H-6), 4.36 (t, 1H, O-H, *J* = 5.32 Hz), 4.99 (brs, 2H, N-H), 5.51-5.67 (m, 2H, H-3, H-4), 6.50 (d, 1H, N-H, *J* = 7.96 Hz), 7.64 (s, 1H, Ar-H), ^13^C-NMR (DMSO, 100 MHz): *δ* (ppm) = 28.7, 32.3, 41.3, 48.3, 62.9, 124.2, 125.5, 127.3, 137.7, 146.4, 152.7, MS (ES, pos) *m*/*z* = 255 (M + 1), 257 (M + 3).



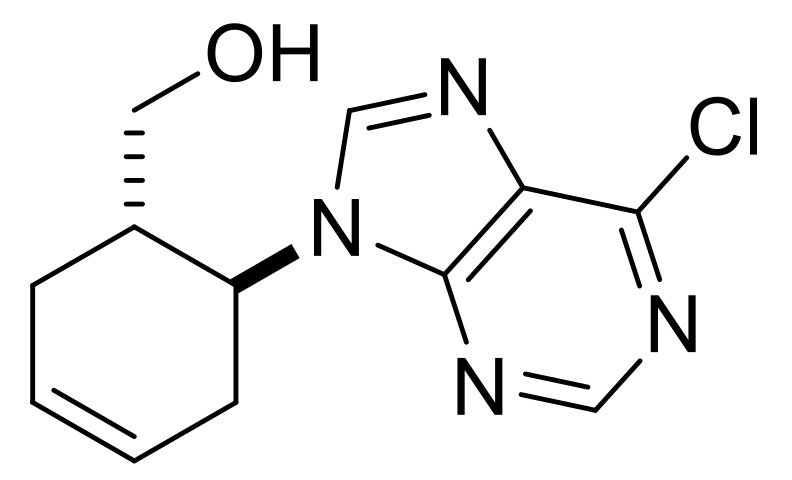

((1*S**,6*S**)-6-(6-Chloro-9H-purin-9-yl)cyclohex-3-en-1-yl)methanol, **(±)-47**.

White solid, m.p. 160–162 °C, 70%; ^1^H-NMR (DMSO, 500 MHz): *δ* (ppm) = 2.13–2.29 (m, 2H, CH_2_), 2.42–2.49 (m, 1H, CH_2_), 2.53–2.62 (m, 1H, H-1), 2.83–2.95 (m, 1H, CH_2_), 2.98–3.05 (m, 1H, OCH_2_), 3.11–3.17 (m, 1H, OCH_2_), 4.69–4.78 (m, 1H, H-6), 5.68–5.85 (m, 2H, H-3, H-4), 8.73–8.78 (m, 2H, Ar-H), ^13^C-NMR (DMSO, 126 MHz): 28.5, 31.3, 39.0, 54.1, 61.7, 124.6, 127.2, 131.6, 147.6, 149.4, 151.7, 152.4; MS (ES, pos) *m*/*z* = 265 (M + 1), 267 (M + 3).

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
