# Peer review of "Alicyclic β- and γ-Amino Acids: Useful Scaffolds for the Stereocontrolled Access to Amino Acid-Based Carbocyclic Nucleoside Analogs"

_molecules, 2019, doi:10.3390/molecules24010161_

Reviewer 1 Report

The authors describe in this manuscript a concise approach to carbocyclic amino-acid nucleoside analogues. The chemistry described is quite simple, and open access to the preparation of a series of nucleoside analogues.

In my opinion, the manuscript coud  be accepted for publication in Molecules.

However, I would suggest to the authors to include some comments on the potential transformation of the racemic compounds into enantiomerically pure derivatives, owing to their interest. Have the authors made any attempt to carry out the enantiomeric separation?

 Overall, the manuscript is well-written and it could be accepted in is present form.

Author Response

Replies to the comments of the Reviewers (Manuscript ID: molecules-414548, Alicyclic β- and γ-amino acids: useful scaffolds for the stereocontrolled access to amino acid-based carbocyclic nucleoside analogues)

Dear Editor,

Please find attached the revised version of the manuscript “Alicyclic β- and γ-amino acids: useful scaffolds for the stereocontrolled access to amino acid-based carbocyclic nucleoside analogues” (Manuscript ID: molecules-414548) submitted to Molecules.

The comments of the Reviewers have been practically accepted and their queries have been answered, their suggestion being included in the manuscript. Our answers are indicated below in blue colour letters. We highly appreciate the Reviewer’s suggestions, which have helped us to improve the manuscript and we hope that the revised version will be considered acceptable for publication in Molecules. All the required modifications made in the manuscript have been highlighted with a yellow background.

Answers to the comments of the Reviewers:

Reviewer 1:

“However, I would suggest to the authors to include some comments on the potential transformation of the racemic compounds into enantiomerically pure derivatives, owing to their interest. Have the authors made any attempt to carry out the enantiomeric separation?”

The synthesis of enantiomerically pure substances is planned to be performed in the framework of a new research project. Based on our earlier experiences in enzymatic resolution involving lactam ring opening by hydrolysis in the presence of Candid antarctica lipase, the access to enantiopure lactam can be achieved. Since the integrity of the stereocenters during the synthetic pathway is not affected the enantiomerically pure target molecules may be accomplished. A relevant sentence in the Conclusion and Outlook part have been added with some references.

“Furthermore, based on our experiences in enzymatic resolution of various bicyclic β- and γ-lactams [23], synthesis of enantiomerically pure substances will be performed.

23. a) Forró, E.; Fülöp, F. Enzymatic Method for the Synthesis of Blockbuster Drug Intermediates –Synthesis of Five-Membered Cyclic γ-Amino Acid and γ-Lactam Enantiomers, Eur. J. Org. Chem. 2008, 5263-5268; b) Forró, E.; Kiss, L.; Árva, J.; Fülöp, F. Efficient Enzymatic Routes for the Synthesis of New Eight-membered Cyclic beta-Amino Acid and beta-Lactam Enantiomers, Molecules 2018, 22, 2221.”

Reviewer 2:

“The reviewer suppose that this is the reason the Authors do not take care to the lack of enantioselectivity in the procedure (why do not add some references about the resolution of the racemic mixture usually obtained?).”

Although the resolution of the racemic mixture of the target compounds have not been explored in the current paper, based on our earlier investigations on the lactam ring opening through enzymatic resolution or by enzymatic ester group hydrolysis it is obvious to plan similar strategies towards the preparation of enantiopure materials.

Relevant references have been added, as well as a related sentence have been incorporated into the manuscript.

Forró, E.; Megyesi, R.; Paál, T. A.; Fülöp, F. Efficient dynamic kinetic resolution method for the synthesis of enantiopure 6-hydroxy- and 6-methoxy-1,2,3,4-tetrahydroisoquinoline-1-carboxylic acid, Tetrahedron: Asymmetry 2016, 27, 1213–1216.

As both reviewers have rated the results presented in our manuscript as with important relevance and indicated that our manuscript can be accepted in Molecules after the revisions, we hope that the corrected version, together with the additional comments provided in this letter, is now suitable for publication.

Sincerely yours,

Lorand Kiss

Kiss Loránd

professor of chemistry

University of Szeged

Institute of Pharmaceutical Chemistry

6720 Szeged, Eötvös u. 6

Tel: +36-62-545561

Tel: +36-30-8904092

E-mail: kiss.lorand@pharm.u-szeged.hu

E-mail: kiss.lorand00@gmail.com

http://www2.pharm.u-szeged.hu/gyki

Hungary

Reviewer 2 Report

The Authors disclosed an interesting approach to the development of a little library of new carbocyclic nucleoside analogues featured by a ring olefin conjugated to a β-amino acid, γ-amino acid or γ-amino alcohol moiety. The synthetic pathway described ,indeed, is inspired by the widespread literature on the use of the Vince lactam.

The reviewer suppose that this is the reason the Authors do not take care to the lack of enantioselectivity in the procedure (why do not add some references about the resolution of the recemic mixture usually obtained?). Otherwise, is not clear how the obtained products could be useful for a biological assay.

However, the manuscript is technically sound, and the data support the conclusions, so even if all procedures applied do not involve any original steps, nevertheless the manuscript is interesting enough for a significant number of readers. Therefore, I am pleased to recommend publication in Molecules.

Author Response

(The authors gave the same response as above.)
